# TACS-Guided Self-Alignment of LVLMs for Explainable Chest X-ray Analysis

## Abstract

Large vision–language models (LVLMs) hold promise for medical imaging but face two critical challenges: dependence on curated human-annotated datasets for alignment and poor robustness to real-world perturbations. We show that LVLMs can produce inconsistent outputs between original chest X-rays and WhatsApp-compressed versions that appear visually indistinguishable. Such failures raise serious concerns for mHealth platforms, where compressed or perturbed images are common in real-world diagnostic workflows. Moreover, current LVLMs often attribute lung abnormality predictions to irrelevant regions outside the lungs—a phenomenon termed out-of-lung saliency (OLS)—which is exacerbated by compression artifacts. These challenges highlight the urgent need for robust and explainable LVLMs in CXR diagnosis.

To address these issues, we propose Self-CXRAlign, a self-alignment framework that enhances explainability robustness through multi-task learning (MTL)-driven supervised fine-tuning (SFT). Self-CXRAlign enforces explainability robustness, ensuring stability of predictions and attributions across original and perturbed images. Central to our method is the Inter-Task Attribution Conflict Score (TACS), a novel metric that guides the selection of auxiliary tasks to reduce attribution conflicts and mitigate negative transfer. By steering SFT with TACS, Self-CXRAlign achieves up to 30% reduction in OLS compared to naïve MTL, paving the way for explainable and trustworthy LVLM deployment in mHealth-driven chest X-ray analysis.

## 1 Introduction

Recent advancements in aligning Large Language Models (LLMs) to generate safe, reliable and trustworthy outputs have become a major research focus, as LLM's are generating significant interest across various sectors such as medicine (Haltaufderheide & Ranisch (2024)), law (Guha et al. (2023); Colombo et al. (2024)), and education (Liu et al. (2024); Ni et al. (2024)). Prominent alignment strategies, such as Supervised Fine-Tuning (SFT) (Ji et al. (2024)) and Reinforcement Learning with Human Feedback (RLHF) (Yao et al. (2023)), rely on extensive human-annotated data to align model outputs with human preferences. However, curating such data is expensive and time-consuming, motivating a shift toward self-alignment—a paradigm where models use their own feedback signals to steer their behavior without external human input. While self-alignment has shown promise in refining text outputs (Madaan et al. (2023)), improving code generation (Wei et al. (2024)), and enforcing behavioral principles in LLMs (Sun et al. (2023b)), its application to Large Vision-Language Models (LVLMs), particularly in medical imaging, remains underexplored.

Self-alignment holds significant potential for medical LVLMs, where data collection is expensive and requires expert annotation. Several powerful medical LVLMs, such as CheXagent (Chen et al. (2024b)), RadFM (Wu et al. (2025)), BiomedGPT (Zhang et al. (2023a)), LLaVA-Rad (Zambrano Chaves et al. (2025)) and XrayGPT (Thawkar et al. (2023)) have been developed for Chest X-Ray (CXR) analysis tasks like report generation and visual question answering. These systems hold great potential for mHealth applications, to enable diagnostic support across geographic and infrastructure constraints, especially in low- and middle-income countries (LMICs) (Ntja et al. (2022); Kalyanpur (2024); Saini et al. (2024)). For example, in India there is only one radiologist per 100,000 people (Arora (2014)), making automated radiology solutions crucial. With widespread use of instant messaging applications such as WhatsApp (with 535M users in India, 118.5M in

Brazil, and >97% penetration in African nations; (GSMA (2022)), a practical mHealth workflow could involve transmitting compressed CXRs alongside textual queries through a low-bandwidth messaging app and receiving an automated report from an LVLM (Ntja et al. (2022); Kalyanpur (2024); Saini et al. (2024)).

However, real-world applications, where compressed or perturbed images are common in CXR diagnostic workflows, introduces certain challenges. Experiments reveal that LVLM's integrated with low-bandwidth messaging applications such as WhatsApp are susceptible to challenges related to prediction and explanation. The prediction instability occurs when the model generate different diagnosis between original and WhatsApp images, although the images are visually indistinguishable. Prior studies in CXR analysis suggest that model predictions are often confounded by spurious correlations, such as features outside the lungs, leading to a phenomenon termed out-of-lung saliency (OLS) (DeGrave et al. (2021); Geirhos et al. (2020); Antony et al. (2025)). This issue, typically revealed by attribution methods such as saliency maps, also extends to LVLMs. Moreover, the problem of OLS is exacerbated under WhatsApp compression, as illustrated in Figure 1, 5.

To address these challenges, we propose Self-CXRAlign, a novel pipeline for self-alignment of LVLM to achieve consistent predictions and explanations across original and perturbed CXR images. Although alignment via RLHF is proposed for LVLM's, it requires carefully curated, large-scale human-annotated dataset (Sun et al. (2023a); Yu et al. (2024)), which are difficult to obtain in medical applications. Instead, Self-CXRAlign adopts a self-alignment approach with Supervised Fine-Tuning (SFT) that does not require additional annotated data. Our framework leverages Multi-Task Learning (MTL) to exploit task relationships and select auxiliary tasks that improve robustness. A key challenge is that naïve MTL can lead to negative transfer (Zhu et al. (2024)), where auxiliary tasks degrade performance on the primary diagnostic task. Exhaustively searching all possible task combinations is computationally infeasible. Our main contribution is the Inter-Task Attribution Conflict Score (TACS), a novel measure of task transferability. By computing TACS with a surrogate model, we formalize conditions to identify auxiliary tasks that mitigate negative transfer and use them for self-aligning LVLM via SFT.

**Contributions:** Following are the key contributions of this paper.

- **Self-CXRAlign**: a novel pipeline for self-alignment of LVLMs in CXR analysis, robust to image perturbations.

- **Attribution Vulnerability of MTL:** Identification of prediction and explanation instabilities in LVLMs, including out-of-lung saliency (OLS).

- **Task Attribution Conflict Score (TACS):** TACS to quantify task transferability in MTL. Formalization of conditions where auxiliary tasks reduce negative transfer, enabling self-alignment via SFT.

- **Empirical gains:** TACS-guided self-alignment yields up to 30% reduction in OLS and 45% improvement in report generation over SOTA LVLMs.

## 2 RELATED WORKS

**Medical foundation models.** Recent advances in large vision-language models (LVLMs) have enabled the development of medical foundation models trained on paired medical image–text datasets. For instance, Med-Flamingo (Moor et al. (2023)) extends OpenFlamingo (Awadalla et al. (2023)) with interleaved medical training data, while models such as CheXagent (Chen et al. (2024b)), LLaVA-Rad (Zambrano Chaves et al. (2025)) and XrayGPT (Thawkar et al. (2023)) fine-tune LVLMs on chest X-ray (CXR) datasets such as MIMIC-CXR to support radiology report generation. Broader medical generalist models, including RadFM (Wu et al. (2025)), BiomedGPT (Zhang et al. (2023a)), LLaVA-Med (Li et al. (2023)) and Uni-Med (Zhu et al. (2024)) curates a medical multi-modal dataset spanning modalities including CXR, CT, and MRI. However, these efforts still require further fine-tuning on task-specific data to be effective for downstream clinical applications.

**Self Alignment of LLM's.** The alignment of LLMs with human intentions and values has recently gained significant attention (Ji et al. (2023)). Supervised fine-tuning (SFT) and reinforcement learning from human feedback (RLHF) have been widely adopted to incorporate human preferences

into model training (Ouyang et al. (2022); Taori et al. (2023); Ji et al. (2024)). Despite their effectiveness, these techniques rely on extensive expert annotations, which are costly and challenging in specialized domains such as medicine (Rafailov et al. (2023); Yuan et al. (2023); Song et al. (2024)). Recent research efforts have also extended RLHF to multimodal scenarios (Sun et al. (2023a); Yu et al. (2024)). Robust alignment variants, such as conservative DPO (cDPO) (Mitchell), robust DPO (rDPO) (Chowdhury et al. (2024)) and PerpCorrect (Kong et al. (2024a)), address the issues related to noisy preferences. In the context of LLM-based medical evaluation, (Zheng et al. (2025)) employ specialized expert models for individual tasks and incorporate reward tokens to optimize all expert models, ensuring better alignment.

Self-alignment methods have been introduced as annotation-efficient alternatives that bypass reliance on human supervision by leveraging self-generated feedback. Early approaches such as Self-Instruct (Wang et al. (2023)) leverages GPT-3 for aligning pretrained language models by generating new instructions and responses for instruction-tuning using its in-context learning capability. Self-CodeAlign (Wei et al. (2024)) introduces a self-alignment pipeline for code LLMs by generating multiple code snippets paired with test-cases for individual tasks and automatically validating them for SFT. Self-Align (Sun et al. (2023b)) follow a principle-driven approach, which is essentially rule-based along with in-context learning for aligning LLM's with human understanding of principles or rules. Other strategies include Self-Debugging(Chen et al. (2024a)), which enables LLMs to debug their predictions via few-shot examples, and self-critiquing (Saunders et al. (2023)), which reveal weaknesses for iterative fine-tuning. These methods demonstrate the potential of self-alignment to enhance performance without costly human annotations, which is particularly appealing for biomedical LVLMs where expert supervision is expensive and scarce.

**Multi-Task Learning for LLM and LVLM** Multi-task learning (MTL) is the paradigm of jointly learning multiple tasks, introducing shared representations that often improve generalization (Evgeniou & Pontil (2004)). In LVLMs, MTL has been applied to enhance in-context visual understanding(Sheng et al. (2024); Chen et al. (2023)), improve visual entity recognition (Caron et al. (2024); Chen et al. (2023)), and manage diverse vision–language tasks using task-specific decoders(Wu et al. (2024)). MTL has also been extended to LVLMs in the medical domain (Zhu et al. (2024)). However, task conflict remains a key challenge, as highlighted by (Kong et al. (2024b); Wu et al. (2024)) where conflicting gradients can lead to negative transfer. MTL has further been leveraged for aligning LLMs via RLHF, addressing naturally occurring differences in individual human preferences (Poddar et al. (2024)). Despite these advances, the application of MTL for self-alignment remains largely unexplored.

## 3 PROBLEM STATEMENT

We study LVLMs for CXR diagnosis, where the model takes an image and a query to generate diagnostic outputs. Our goal is to improve their robustness to perturbations—particularly compression artifacts from low-bandwidth platforms like WhatsApp—that degrade both predictions and explanations. To this end, we propose a multi-task learning (MTL) framework that models inter-task relationships and identifies auxiliary tasks that avoid negative transfer, which can be used for self-alignment, thereby improving LVLM robustness to compression-induced perturbations.

**Notations.** We denote $[n] = \{1, \ldots, n\}$, the set of all positive integers ranging from 1 to $n$. For $x \in \mathbb{R}^d$, denote $\|x\| = \sqrt{x^\top x}$

### 3.1 PROBLEM SET UP

**Large Vision Language Model (LVLM).** An LVLM $f$ processes an image $X_i$ and an instruction $X_t$ to generate a textual output $y$. Architecturally, an LVLM consist of (Liu et al. (2023); Zhang et al. (2023b); Koh et al. (2023)): an image encoder, typically a pretrained Vision Transformer to generate image embedding, $H_i = q(X_i)$; a projection layer that maps visual embeddings into the language embedding space via $I = W_P H_i$; and an autoregressive language decoder, which is an LLM.

Formally, in the language decoder, let the image tokens be $I = \{i_1, .., i_M\}$ and the text tokens be $T = \{t_1, .., t_{j-1}\}$, where the text sequence includes both the query and previously generated

tokens. The combined input at step $j$ is: $Z = [i_1, .., i_M, t_1, .., t_{j-1}] \in \mathbb{R}^{(M+j-1) \times d}$, where $d$ is the hidden dimension. The tokens pass through multiple transformer layers, consisting of attention (MSA) and feedforward layers (MLP). Each transformer layer computes $Q = ZW_Q$, $K = ZW_K$ and $V = ZW_V$, where $W_Q, W_K, W_V \in \mathbb{R}^{d \times d_k}$. The scaled dot-product attention is:

$$A = softmax(\frac{QK^T}{\sqrt{d_k}}) \in \mathbb{R}^{(M+j-1) \times (M+j-1)} \tag{1}$$

Instruction tuning aligns the LLM to predict answers auto-regressively. $p(y \mid X_i, X_t) = \prod_{j=1}^{L} p_\theta(y_j \mid X_i, X_t, y_{<j})$, for a sequence of length $L$ where $\theta$ denotes trainable parameters.

In the CXR diagnosis setting, the LVLM is trained on instructions such as: *Describe the lung abnormalities in the CXR image*, producing a multi-line radiology-style report, or *List the names of lung abnormalities in the CXR image*, producing a set of abnormality labels.

**Explainability in LVLM.** To study robustness, we analyze LVLMs both in terms of prediction and explanation instability. We focus on the instruction - *List the names of lung abnormalities in the CXR image*. For this instruction, the LVLM is trained with a tokenizer where each lung abnormality *(e.g., Atelectasis, Cardiomegaly)* corresponds to a unique token, and the model is instruction-tuned to generate these tokens sequentially. To analyze explanations, we consider two attribution techniques - saliency maps and attention maps.

**Definition 1.** *Saliency Map (SM). Given an LVLM $f$ with input image $X_i$, instruction $X_t$ and previously generated text $y_{<j}$, the saliency map of $X_i$ for the token generated at step $j$ is defined as $g(x) = \nabla_{X_i} f_j(X_i, X_t, y_{<j})$*

During inference, when the model generates a token corresponding to a lung abnormality, the associated saliency map is generated using Definition 1. Examples of such maps are shown in Figure 1, with a schematic overview provided in the Appendix.

*Attention maps:* For each predicted token, the image and text tokens pass through multiple layers of MSA and MLP. We extract attention maps (Equation 1) at every layer and head. Since each attention map contains weights for both image and text tokens, we retain only the image-token attention for analysis (see Figure 5 in Appendix).

The attributions generated using saliency map and attention maps reveal how much the generated token relies on different regions of the input CXR.

## 3.2 Challenges in Diagnosis of CXR Images over WhatsApp

Transmitting CXR images over low-bandwidth platforms such as WhatsApp introduces compression artifacts that degrade model reliability. State-of-the-art (SOTA) models for CXR diagnosis face two key issues (Antony et al. (2023)):

**Prediction Instability Problem (PIP).** PIP of a predictive model is defined as the probability of disagreement between the predictions on a randomly perturbed instance and the true instance. PIP is evaluated using *PI Score* (please refer Appendix).

**Out-of-Lung Saliency (OLS).** refers to models identifying regions outside the lungs as significant contributors to predictions of lung abnormalities, as measured by OLS Score (please refer Appendix). A high *OLS* suggests that the majority of the dataset is affected by OLS, while a low score implies lesser impact.

Our evaluation of SOTA vision models (ViTs, ResNet-50) and pretrained medical LVLMs (Rad-DINO, CheXagent) on both original and WhatsApp-compressed MIMIC-CXR test images shows that PIP can reach 30%, while OLS is further exacerbated under compression ((Table 3b, Figure 3a). These results highlight the vulnerability of LVLMs to prediction and explanation degradation in perturbed settings. To address this, we propose an MTL-based self-alignment framework that improves robustness to such perturbations, particularly in low-data diagnostic scenarios.

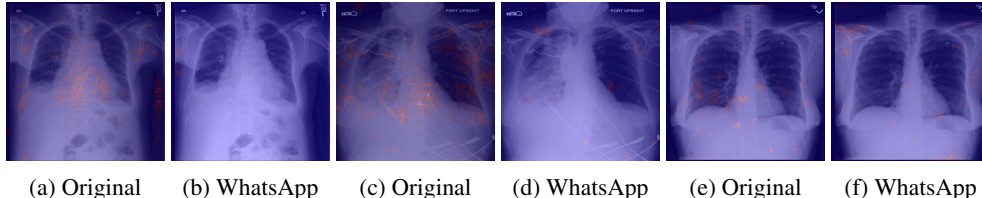

| (a) Original | (b) WhatsApp | (c) Original | (d) WhatsApp | (e) Original | (f) WhatsApp |

Figure 1: Saliency-based localization of lung abnormalities in LLaVA-CXR for original vs. WhatsApp-compressed CXR images. Despite visual similarity, the localizations differ markedly.

### 3.3 ATTRIBUTION VULNERABILITY IN MTL

**MTL:** Multi-label problems such as those of CXR diagnosis have been effectively addressed in recent studies using MTL (Ghamizi et al. (2023); Fifty et al. (2021); Chen et al. (2018)). In MTL setting the goal is to learn a mapping from input space to logits, $f : \mathbb{R}^d \to \mathbb{R}^{\mathcal{T}}$ parameterized by $\theta$, where $[\mathcal{T}]$ denote the set of tasks. The task-specific model, often denoted by $f_t$ is given by $f_t(X, \theta)$, yielding predictions $\hat{y}_t = \mathbf{1}\{f_t(X) > 0.5\}$. The per-task loss is $L_t : \mathbb{R} \times \mathbb{R} \to \mathbb{R}$ and the standard MTL objective is $\min_\theta [L_{MTL}(\theta) := \sum_{t \in [\mathcal{T}]} L_t(f_t(\theta, X), Y)]$.

To study attribution robustness in MTL, we begin by defining attribution vulnerability in this setting. For a task $t$, the attribution map $g_t(x; f_t)$ (Tan & Tian (2023)), measures the importance of input $x$ towards predicting $\hat{y}_t = f_t(x)$. Since the notion of attribution map for MTL does not exist in literature we propose the following definition of aggregate attribution map for MTL as follows.

$$g(x) = \frac{1}{\mathcal{T}} \sum_{i=1}^{\mathcal{T}} g_t(x) \qquad (\text{ATT})$$

where $g_t(x, f)$ is written as $g_t(x)$ for brevity and $\mathcal{T}$ is the number of tasks. This choice is governed by simplicity and one could potentially think of otherways to aggregate the task attribution. Next we define the vulnerability of such a map.

**Definition 2.** *(Attribution Vulnerability for MTL Network). Given an MTL model $f$ consisting of $\mathcal{T}$ tasks, the attribution vulnerability of $f$ is given*

$$AV([\mathcal{T}])^2 = \frac{1}{[\mathcal{T}]} \mathbb{E}_{x,x'} \|v(x, x')\|^2, \quad v(x, x') = \sum_{t=1}^{\mathcal{T}} v_t \qquad (\text{Vulnerability})$$

*where $v_t(x, x') = g_t(x) - g_t(x')$ is the vulnerability vector for task $t$ and $x'$ is the perturbed version of $x$.*

If $AV$ is high then the explanations from the associated model will be vulnerable to distortions in the datapoint $x$. On the other hand if $AV$ is low the explanations offered will be more robust. In practice, we use saliency maps $g_t(x) = \nabla_x f_t(x)$ as the attribution method to understand which parts of an input image were most important for the model's specific prediction (Tan & Tian (2023)).

In the setting of the paper one wishes to build MTL models which offer robust explanations, i.e. explanations which are robust to the perturbations of data, for a focused task. In the sequel we will refer this as primary task and rest of the tasks as auxiliary tasks.

Consider a primary task $p$ and $[\mathcal{T}]$ be the set of auxiliary tasks. One can ask the following questions *(a) Does training with all tasks improve attribution robustness for the primary task?* and *(b) If not, can we identify subsets of auxiliary tasks that yield more robust explanations?*

Let $AV_i$ denote $AV(\{i\})$ and $AV_{ij} = AV(\{i, j\})$ for all $i, j \in [\mathcal{T}]$. Computation of $AV_{ij}$ requires training a model with tasks $i$ and $j$, while computation of $AV_i$ requires training a model with only task $i$. If $AV_i > AV_{ij}$ then it can deemed that training $\{i, j\}$ together has reduced the attribution vulnerability of task $i$, i.e. explanations of task $i$ will be more robust to data perturbations. Using this insight we rephrase the questions posed above by the following questions.

- Does Vanilla MTL(using all tasks) always help in reducing vulnerability

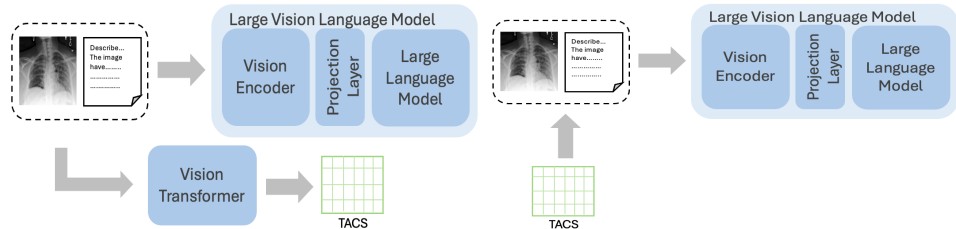

(a) Pre-training of LVLM and Vision Transformer          (b) Self-alignment of LVLM

Figure 2: Self-CXRAlign pipeline. (a) Pre-training the LVLM with image–text pairs while simultaneously training a ViT with image–label pairs to compute TACS. (b) Self-alignment of the LVLM using auxiliary tasks selected based on TACS.

- Are there subsets of Tasks which can yield more robust explanations and can they be easily identified without brute force search

In the sequel, we attempt to answer the above two questions, which essentially characterizes the increase /decrease in attribution robustness when a given primary task is grouped with a certain set of auxiliary tasks. Using this insight, a pipeline for self-alignment of LVLM for robust explanations will be developed for CXR diagnosis.

## 4    SELF-CXRALIGN: TACS-GUIDED PIPELINE FOR SELF-ALIGNMENT

Self-alignment is a paradigm where a model leverages its own existing capabilities or automatically generated data to improve its performance. Towards this end, we derive conditions for characterizing attributional vulnerability of MTL. Then, we formalize conditions for identifying auxiliary tasks that mitigate negative transfer. Furthermore, we describe how these insights are leveraged for self-alignment of LVLM using surrogate model which in our case is a Vision transformer (ViT).

To aid the study, we introduce the following definition

**Definition 3.** *(Task Attribution Conflict Score (TACS)). TACS between two tasks $i, j$ is defines as*

$$\rho_{i,j} = \frac{\mathbb{E}(v_i^T v_j)}{\sqrt{\mathbb{E}(\|v_i\|^2)}\sqrt{\mathbb{E}(\|v_j\|^2)}} \tag{2}$$

*where $v_i$ and $v_j$ are attribution vulnerability vectors of tasks $i$ and $j$ respectively when a model is jointly trained with tasks $\{i, j\}$.*

Computation of $AV_i$ and $AV_j$ requires training the model with tasks $i$ and $j$, respectively, while $AV_{ij}$ requires training the model with both tasks $\{i, j\}$. We observe through extensive experimentation that $\rho_{ij}$ is roughly similar if $v_i$ and $v_j$ are obtained individually from task $i$ and task $j$ respectively, i.e. computation of $\rho$ is transferrable across models trained jointly and singly. We characterize this observation via the following assumption.

**Assumption 1.** *(Transferability Assumption.) For any set of Tasks $[\mathcal{T}]$, for all $i, j \in [\mathcal{T}]$,*

$$AV^2([T]) = \frac{1}{\mathcal{T}} \sum_{i=1}^{\mathcal{T}} (AV_i^2 + \sum_{ij} \rho_{ij} AV_i AV_j)$$

*holds.*

To understand the importance of TACS consider the following proposition

**Proposition 1.** *(Task Selection) Let $p, q$ be any two tasks. The Attribution Vulnerability score of multitask model $f = [f_p, f_q]^\top$ be $AV_{pq}$ while $AV_p$ and $AV_q$ be the attribution vulnerability score for tasks $p$ and $q$ respectively. Under the transferability assumption if $AV_{pq} < AV_p$ then $\rho_{pq} < 0$*

The proof is presented in Appendix. This clearly suggests that TACS serve as a necessary condition for lowering the Attributional vulnerability. It further gives insight that when the attributions from

the tasks $p$ and $q$ are in conflict, i.e. $\rho_{pq} < 0$, the attributional vulnerability is reduced. This is indeed counter-intuitive that tasks which are in some sense conflicting can provide better explainability when trained together.

This immediately suggests that training with MTL incorporating all tasks often may not provide explanations which are robust to vulnerability. For sake of exposition we make the following assumption that all single tasks are *equally vulnerable*, $AV_q = 1$ and provide the following proposition for consideration.

**Proposition 2.** *(Attribution Vulnerability of MTL) Consider an MTL model $f$ with primary task $p$ and a set of auxiliary tasks $[\mathcal{T}]$ each distinct from $p$. Let $AV_q = 1, \forall q \in [\mathcal{T}]$. Under the transferability assumption if $\sum_{q=1}^{\mathcal{T}} \rho_{pq} > 0$, then $AV(\{p \cup [\mathcal{T}]\}) > AV_p$.*

This points to the issue that if the sum of the TACS score is positive then models obtained from vanilla MTL(training all tasks together) could be more vulnerable to noise perturbations. It is thus important to seek subsets of tasks which can yield models with reduced attributional vulnerablility. However such subsets may not exist in the first place. To this end consider the proposition which can be considered as a corollary to the previous proposition.

**Proposition 3.** *(Non-existence of Subsets) Consider an MTL models $f$ with primary task $p$ and a set of auxiliary tasks $[\mathcal{T}]$. For any $\mathcal{S} \subseteq [\mathcal{T}]$, $AV(\{p \cup \mathcal{S}\}) > AV_p$ if $\rho_{pt} > 0$ for all $t \in [T]$ and $\rho_{ts} > 0$ for all $t, s \in \mathcal{S}$.*

Though the above result is negative but if the tasks are more or less conflicting in nature there could be subsets which can reduce vulnerability. Identification of such subsets without brute force searching over all possible task combinations is a challenging task. We present the following necessary condition which will subsequently aid in identifying good subsets which reduces Attribution Vulnerability.

**Proposition 4.** *(Task subset selection) Consider an MTL models $f$ with primary task $p$ and a set of auxiliary tasks $\mathcal{S} \subseteq [\mathcal{T}]$. Under the transferability assumption if $AV(\{p \cup \mathcal{S}\}) < AV_p$ then there must exist at least one $q \in [\mathcal{S}]$ such that $\rho_{pq} < 0$.*

The proof of Propositions 1-4 are presented in Appendix. From propositon 4, given a primary task $p$ and set of auxiliary tasks $[\mathcal{T}]$, we can find a task subset $\mathcal{S} \subseteq [\mathcal{T}]$ such that $AV_{\{p\} \cup \mathcal{S}} < AV_{[\mathcal{T}]}$, where $[\mathcal{T}]$ is the MTL formed by adding all the auxiliary tasks in $[\mathcal{T}]$ to the primary task $p$. Based on this observation we propose the approach for self-alignment in LVLM.

### 4.1 SELF-CXRALIGN PROCEDURE

The schematic diagram showing Self-CXRAlign is shown in Figure 2. The procedure consists of three main steps:

*(i) TACS computation with Surrogate:* A smaller surrogate model - Vision Transformer, is trained on the primary and auxiliary tasks in $p \cup [\mathcal{T}]$. For the primary task $p$, TACS values $\rho_{p,q}$ (Eq.2) are computed for each auxiliary task $q \in [\mathcal{T}]$, using saliency map generated for original and WhatsApp compressed (perturbed) images.
*(ii) Self-Generated Curriculum:* Auxiliary tasks with $\rho_{p,q} < 0$ are automatically selected, as they are known to mitigate negative transfer and improve robustness for the primary task (Proposition4).
*(iii) Self-Improvement:* The base LVLM is fine-tuned using Supervised Fine-Tuning (SFT) on the primary together with the selected auxiliary tasks, $\mathcal{S} = \{q : \rho_{p,q} < 0, q \in [\mathcal{T}]\}$.

This pipeline enables the model to align its internal representations to be more robust to perturbations using a curriculum derived from its own analysis. Unlike prior self-alignment strategies based on human-defined rules or iterative fine-tuning, Self-CXRAlign leverages task relationships in MTL to mitigate negative transfer and enhance attributional robustness.

## 5 EXPERIMENTS

The experiments are designed to demonstrate: (i) the susceptibility of SOTA models to PIP and OLS (ii) the selection of auxiliary task (iii) the robustness improvements in LVLM explanations

via TACS-guided self-alignment (iv) the performance gains in report generation by TACS-guided self-alignment; and (v) the transferability of TACS-selected tasks.

## 5.1 EXPERIMENT SETTINGS

**Datasets and Evaluation.** We use the MIMIC-CXR (Johnson et al. (2019)), which contains CXR images paired with radiology reports and is widely used for LVLM training. The dataset consists of 377,110 CXR training images and 5131 test images, covering 14 lung abnormalities, where each image may contain multiple abnormalities. To study robustness under real-world perturbations, we create a WhatsApp-CXR dataset by transmitting the MIMIC test images over WhatsApp in an automated manner. The original 5131 test images occupy 8.34 GB, while the WhatsApp-compressed versions occupy only 1.22 GB. Further details are provided in the Appendix. Training and self-alignment are performed on the MIMIC-CXR training set, and evaluation is conducted on both the original and WhatsApp-compressed test sets. The baselines for report generation are SOTA pre-trained LVLM's trained on CXR and multi-modal medical data. Generated reports are evaluated for factual accuracy and lexical similarity using standard metrics: BLEU, ROUGE, and RadGraph-based F1 scores (Jain et al. (2021)).

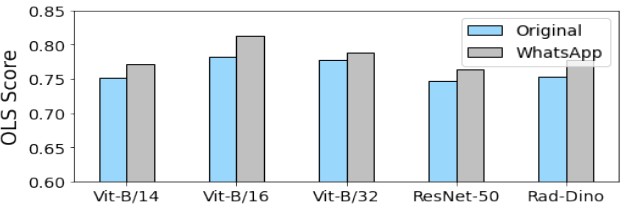

| Model | PIP Score |
|---|---|
| ViT-B/14 | 4.47 |
| ViT-B/16 | 6.20 |
| ViT-B/32 | 5.18 |
| ResNet-50 | 6.28 |
| Rad-Dino | 30.51 |
| CheXagent | 11.57 |

(a) OLS Score results of SOTA models  (b) PIP Score Results of SOTA models

Figure 3: OLS Score and PIP Score Results of SOTA models on MIMIC-CXR dataset. From the results, it can be seen that SOTA models are susceptible to OLS and PIP challenges

**LVLM Architecture.** We adapt LLaVa Liu et al. (2023) with two modifications -1)the open-sourced vision encoder CLIP in LLAVa is replaced by BiomedCLIP-CXR (Zhang et al. (2023c)) and 2) a domain-specific tokenizer and vocabulary for language Model. We adapt LLaMA2 Vicuna-7B-v1.5 (Touvron et al. (2023)) as language Model, as in LLaVa. The resulting model, LLaVA-CXR is trained with an input image resolution is $518 \times 518$. LLaVA-CXR and ViT training details are provided in Appendix.

## 5.2 EXPERIMENT RESULTS

**Demonstration of prediction and explanation challenges in SOTA models.** We evaluate SOTA vision models (ViT-B/14, ViT-B/16, ViT-B/32, ResNet-50) and pretrained LVLMs (Rad-DINO, CheXagent) on original and WhatsApp-compressed test images. Results show that these models suffer from PIP and degraded explainability. As shown in Table 3b, all models exhibit PIP up to 30%,

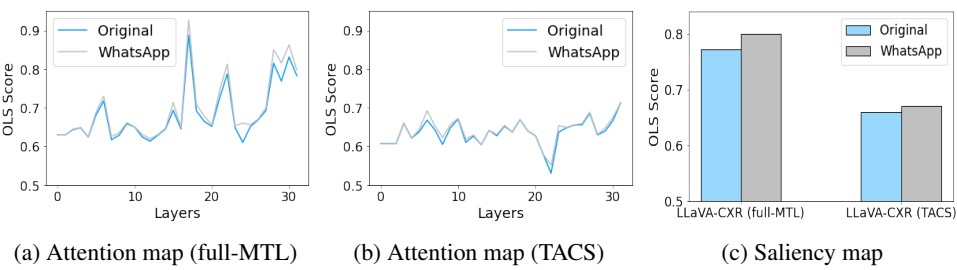

(a) Attention map (full-MTL)  (b) Attention map (TACS)  (c) Saliency map

Figure 4: (a–b) OLS scores from attention maps across layers of LLaVA-CXR (full-MTL) and LLaVA-CXR (TACS), evaluated on original and WhatsApp-compressed images. (c) OLS scores from saliency maps for both models, showing that LLaVA-CXR (TACS) effectively reduces OLS.

| Model | Rogue-1 | | Rogue-L | | BLEU-1 | | BLEU-4 | | Rad-graph | |
|---|---|---|---|---|---|---|---|---|---|---|
| | Orig | WA | Orig | WA | Orig | WA | Orig | WA | Orig | WA |
| Biomed-GPT | 08.85 | 08.22 | 06.97 | 06.56 | 05.46 | 05.21 | 00.67 | 00.57 | 03.89 | 02.88 |
| Rad-FM | 13.64 | 12.98 | 10.01 | 09.20 | 09.43 | 08.89 | 01.22 | 01.06 | 05.14 | 04.73 |
| LLaVA-Med | 18.26 | 17.83 | 12.61 | 11.59 | 11.93 | 10.77 | 00.95 | 00.85 | 08.32 | 08.00 |
| XrayGPT | 13.63 | 12.50 | 10.88 | 10.13 | 0.10 | 09.27 | 01.09 | 00.89 | 05.02 | 03.68 |
| LLaVA-CXR(MTL) | 29.83 | 28.75 | 21.45 | 20.33 | 22.67 | 21.40 | 05.35 | 04.88 | 16.59 | 15.25 |
| LLaVA-Rad | 12.15 | 11.02 | 10.88 | 10.64 | 01.08 | 00.99 | 00.11 | 00.10 | 02.00 | 01.90 |
| CheXagent | 30.08 | 29.96 | 20.52 | 19.58 | 22.17 | 22.08 | 04.54 | 04.23 | 18.30 | 17.62 |
| LLaVA-CXR(TACS) | **33.01** | **32.79** | **22.33** | **21.48** | **25.43** | **24.39** | **05.37** | **04.82** | **19.74** | **18.34** |

Table 1: Comparison of radiology report for CXR images for the primary task in MIMIC-CXR original and WhatsApp test images, generated by SOTA LVLM, LLaVA-CXR(full-MTL) and LLaVA-CXR(TACS). The results show that LLaVA-CXR(TACS) outperforms the baselines

while average OLS is 76% for original images and increases to 79% for WhatsApp-compressed images (Figure 3a).

**Selection of auxiliary tasks.** We choose Fracture as the primary task, since it has the fewest samples. Training on this task alone degrades LVLM report quality, necessitating the use of auxiliary tasks to mitigate negative transfer. We therefore train two variants of our model- LLaVA-CXR(full-MTL), trained using all tasks in MIMIC-CXR and LLaVA-CXR(TACS), trained using auxiliary tasks identified via TACS. There are $2^{13}$ possible combinations of auxiliary tasks that can be grouped with the primary tasks. Using TACS (Equation 2) computed on Vit-B/14 trained using MIMIC-CXR dataset (Table 4 in Appendix), we identified 8 auxiliary tasks - Atelectasis, Cardiomegaly, Consolidation, Edema, Enlarged Cardiomediastinum, Support Device, Lung Opacity, and Pleural Effusion.

**Self-CXRAlign improves LVLM explainability robustness.** For the instruction *List all lung abnormalities in the CXR image*, we compute attribution maps (saliency maps, Def. 1; and attention maps, Eq. 1) for both original and compressed test images using LLaVA-CXR(full-MTL) and LLaVA-CXR(TACS). The OLS scores (Eq 5) of attention-map, aggregated layer-wise, are reported in Figures 4a, 4b; and the OLS scores of saliency map are reported in Figure 4c. Results show that LLaVA-CXR(TACS) reduces OLS by up to 30% for attention maps and 15% for saliency maps compared to LLaVA-CXR(full-MTL).

**Self-CXRAlign improves LVLM report generation.** Generated reports from LLaVA-CXR(TACS), LLaVA-CXR(full-MTL), and other SOTA pre-trained LVLMs are evaluated on the primary task (Table 1). Results clearly demonstrate that LLaVA-CXR(TACS) achieves an average of 55% higher ROUGE and BLEU scores compared to SOTA LVLMs, and 12% improvement over LLaVA-CXR(full-MTL). Results of report generated for both primary and auxiliary tasks indicate that LLaVA-CXR(TACS) achieves an average of 40% higher ROUGE and BLEU scores compared to SOTA LVLMs (Table 2 in Appendix). For diagnostic label prediction, LLaVA-CXR(TACS) achieves 10% higher F1 scores compared to SOTA baselines (Table 3 in Appendix).

**Transferability of TACS-based tasks.** We compute TACS scores (Equation 2) on ViT-B/14, ViT-B/16, ViT-B/32 trained on MIMIC-CXR training data using attributions generated by saliency map (Def. 1) for both the original and WhatsApp test images. The TACS score of the primary task remain consistent across the models and are presented in Table 4 in Appendix. Also, TACS score remains stable even when evaluated on only 50% of test set.

Our experiments reveal that SOTA LVLM's suffer from significant prediction instability and out-of-lung saliency. TACS effectively identifies auxiliary tasks that mitigate these issues, improving both attribution robustness and downstream report generation. While TACS is evaluated using saliency map, the attribution robustness of attention map is also getting better. TACS-driven self-alignment of LVLM consistently outperforms SOTA LVLM baselines and LVLM trained by naively adding all the tasks, across original and perturbed test conditions. Our findings suggest that Self-CXRAlign provides a practical pathway to deploying LVLMs in real-world medical settings, where CXR images are often subject to network-induced perturbations and compression artifacts. Furthermore, our approach mitigates negative task transfer without requiring costly additional data curation.

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

# 6 APPENDIX

## 6.1 ATTRIBUTION ROBUSTNESS USING AUXILIRAY TASKS

**Proposition 5.** *(Attribution Vulnerability of MTL) Consider an MTL model $f$ with primary task $p$ and a set of auxillary tasks $[\mathcal{T}]$ each distinct from $p$. Let $AV_q = 1, \forall q \in [\mathcal{T}]$. Under the transferability assumption if $\sum_{q=1}^{\mathcal{T}} \rho_{pq} > 0$ and all $\rho_{qs} > 0, q, s \in [\mathcal{T}]$, then $AV(\{p \cup [\mathcal{T}]\}) > AV_p$.*

*Proof.* From Transferability assumption and using $AV_i = 1$ for all $i \in \{p \cup [\mathcal{T}]$ the following two equations can be inferred by direct computation.

$$AV^2([\mathcal{T}]) = \frac{1}{\mathcal{T}} \left( \sum_{i,j=1}^{\mathcal{T}} \rho_{ij} AV_i AV_j \right) > 1$$

$$AV^2(\{p, [\mathcal{T}]\}) = \frac{1}{\mathcal{T}+1} \left( AV_p^2 + 2 \sum_{q=1}^{\mathcal{T}} \rho_{ps} AV_p AV_s + \mathcal{T} AV^2([\mathcal{T}]) \right)$$

Thus it follows that

$$AV(\{p, [\mathcal{T}]\}) \geq 1 = AV_p$$

$\square$

**Proposition 6.** *(Non-existence of Subsets) Consider an MTL models $f$ with primary task $p$ and a set of auxillary tasks $\mathcal{T}$. For any $\mathcal{S} \subseteq [\mathcal{T}]$, $AV(\{p \cup \mathcal{S}\}) > AV_p$ if $\rho_{pt} > 0$ for all $t \in [T]$ and $\rho_{st} > 0$ for all $s, t \in [\mathcal{T}]$.*

*Proof.* Identical to the proof of the previous proposition. For any subset $\mathcal{S} \subseteq [\mathcal{T}]$

$$AV^2(\mathcal{S}) = \frac{1}{|\mathcal{S}|+1} \left( \sum_{i,j=1}^{|\mathcal{S}|} \rho_{ij} AV_i AV_j \right) > 1$$

$$AV^2(\{p \cup \mathcal{S}\}) = \frac{1}{|\mathcal{S}|+1} \left( AV_p^2 + 2 \sum_{q=1}^{|\mathcal{S}|} \rho_{ps} AV_p AV_s + |\mathcal{S}| AV^2(\mathcal{S}) \right)$$

Thus it follows that

$$AV(\{p \cup \mathcal{S}\}) > 1 = AV_p$$

Hence proved $\qquad \square$

**Proposition 7.** *(Task Selection) Let $p, q$ be any two tasks. The Attribution Vulnerability score of multitask model $f = [f_p, f_q]^\top$ be $AV_{pq}$ while $AV_p$ and $AV_q$ be the attribution vulnerability score for tasks $p$ and $q$ respectively. Under the transferability assumption if $AV_{p,q} < AV_p$ holds then $\rho_{p,q} < 0$.*

*Proof.* By assumption the following holds

$$AV_{pq}^2 = AV_p^2 + AV_q^2 + 2\rho_{pq} AV_p AV_q$$

If $AV_{p,q} \le AV_p$ then $AV_p^2 + AV_q^2 + 2\rho_{pq} AV_p AV_q \le AV_p^2$, which implies that $2\rho_{pq} AV_p AV_q \le -AV_q^2$. Since $AV_p$ and $AV_q$ are both positive, it must be that $\rho_{pq} < 0$. $\qquad \square$

Using Proposition 7, we can identify those auxiliary tasks $q \in [\mathcal{T}]$, which should *not* be augmented with $p$ for improving attribution robustness. The Proposition suggests that those $q \in [\mathcal{T}]$ for which $\rho_{p,q} < 0$ are the auxiliary tasks that can improve attribution robustness.

**Proposition 8.** *(Task subset selection) Consider an MTL models $f$ with primary task $p$ and a set of auxiliary tasks $\mathcal{S} \subseteq [\mathcal{T}]$. Under the transferability assumption if $AV(\{p \cup \mathcal{S}\}) < AV_p$ then there must exist at least one $q \in [\mathcal{S}]$ such that $\rho_{pq} < 0$.*

*Proof.* For any task $p$ and set of tasks $[S]$ which does not include $p$, a consequence of the transferability assumption is as follows $AV(\{p \cup \mathcal{S}\})^2 = AV_p^2 + 2\sum_{q \in [S]} \rho_{pq} AV_p AV_q + AV([\mathcal{S}])$ . Thus $AV(\{p \cup \mathcal{S}\})^2 \le AV_p^2 \implies 2\sum_{q \in [S]} \rho_{pq} AV_p AV_q \le -AV([\mathcal{S}])$. This is only possible if there exist at least one $q \in [\mathcal{S}]$ such that $\rho_{pq} < 0$. $\qquad \square$

From propositon 8, given a primary task $p$ and set of auxillary tasks $[\mathcal{T}]$, we can find a task subset $\mathcal{S} \subseteq [\mathcal{T}]$ such that $AV_{\{p\} \cup \mathcal{S}} < AV_{[\mathcal{T}]}$, where $[\mathcal{T}]$ is the MTL formed by adding all the auxiliary tasks in $[\mathcal{T}]$ to the primary task $p$. Based on this observation we propose the approach for self-alignment in LVLM.

## 6.2 CHALLENGES OF CXR DIAGNOSIS

**Prediction Instability Problem** Prediction Instability of a predictive model is defined as the probability of disagreement between the predictions on a randomly perturbed instance and the true instance. Consider a model $f$ that classifies a CXR image into one or multiple categories. Ideally, M should provide the same prediction for both the original and WhatsApp-compressed CXR image. Consider a dataset $D = \{X_i, \text{R}(X_i)\}_{i=1}^N$ consisting of $N$ pairs of original and WhatsaApp-compressed CXR images denoted by $X_i$, and $\text{R}(X_i)$ respectively.

**Definition 4.** *PI Score The estimate of PI Score due to image perturbation by WhatsaApp can now be defined as the fraction of pairs where the predictions differed expressed as percentage.*

$$PI\,Score(f; D) = \frac{1}{N} \left( \sum_{\{X_i, \text{R}(X_i)\} \in D} PI(X_i, \text{R}(X_i), f) \right) \times 100 \qquad (3)$$

*and $PI(X_i, \text{R}(X_i), f) = \mathbf{I}(y(X_i; f)) \ne \mathbf{I}(y(\mathbf{R}(X_i); f))$*

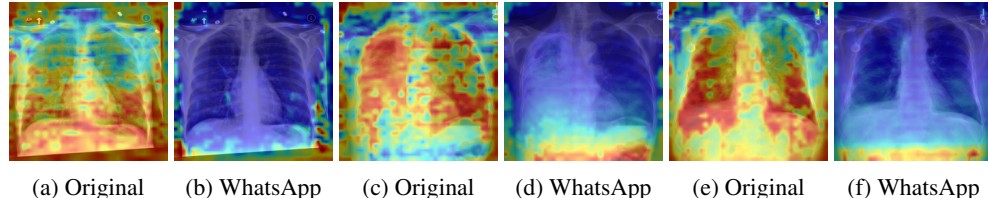

(a) Original    (b) WhatsApp    (c) Original    (d) WhatsApp    (e) Original    (f) WhatsApp

Figure 5: Attention-based localization of lung abnormalities in LLaVA-CXR for original vs. WhatsApp-compressed CXR images. Despite visual similarity, the localizations differ markedly..

**Out-of-Lung Saliency (OLS)** refers to models identifying regions outside the lungs as significant contributors to predictions of lung abnormalities. Clearly such *explainations* are not acceptable. Experiments conducted on MIMIC-CXR (Johnson et al. (2019))—revealed that attribution maps generated by state-of-the-art (SOTA) models trained on these datasets exhibit a high OLS problem. Furthermore, the OLS issue is exacerbated in WhatsApp-compressed images, a trend also observed in (DeGrave et al. (2021); López-Cabrera et al. (2021); Antony et al. (2023)).

**Definition 5.** *OLS Score of a model* $\mathtt{f}$ *on a dataset $D$ is defined as the percentage of images in $D$ with an Intersection Over Lung-Region (IOL) value less than a threshold $\eta$.*

$$OLS\ Score_\eta(D;\mathtt{f}) = \frac{1}{N}\left(\sum_{x \in D} IOL(x;\mathtt{f}) < \eta\right) \times 100$$

$$IOL(x;\mathtt{M}) = \frac{\mathtt{Pix}\left(heatmap\text{-}region(y(x;\mathtt{f})) \cap lung(x)\right)}{\mathtt{Pix}\left(heatmap\text{-}region(y(x;\mathtt{f}))\right)}$$

*where* $\mathtt{Pix}(R(x))$ *counts the number of Pixels in region $R(x)$, a subset of the pixels in the original image $x$.*

The value of $\eta$ is set to 0.4 (Antony et al. (2023)). *IOL* is obtained through a lung segmentation algorithm (Ronneberger et al. (2015)) which aims to segment lungs in a CXR image. The IOL value ranges from 0 to 1, where an IOL of 0 indicates that the prediction relies entirely on pixels outside the lung region, while an IOL of 1 indicates that the prediction is based solely on pixels within the lung region. A high *OLS* suggests that the majority of the dataset is affected by OLS, while a low score implies lesser impact. It was empirically demonstrated (Antony et al. (2023)) that augmenting primary task with auxiliary from a similar dataset can reduce the problem of OLS. However, no justification was provided. In the next section, we attempt to address these challenges using a principled approach that leverage relation between tasks via MTL to improve robustness.

### 6.3 LIMITATIONS AND FUTURE WORK

First, we limit our study to MIMIC-CXR dataset, since it is a datasaet consisting of CXR image, text pairs. In future, it would be beneficial to contribute such CXR image, text datasets and conduct the study for a wider range of lung abnormalities. Secondly, our study focuses on image perturbation caused by WhatsApp based compression. In future it would be beneficial to consider various other types of image perturbations. Additionally, the tasks with negative transfer are filtered out. These negative tasks could be used in a mixture-of-expert loop without hindering robustness. Furthermore, our study is constrained to CXR domain. In future it would be fruitful to self-align LVLM for diverse domains such as satellite imagery, surveillance data, agricultural supervision etc.

### 6.4 TRAINING DETAILS

#### 6.4.1 LVLM ARCHITECTURE AND TRAINING DETAILS

We use LVLM adapted from LLaVa with two changes -1)the open-sourced vision encoder CLIP in LLAVa is replaced by BiomedCLIP-CXR (Zhang et al. (2023c)) and 2) a domain-specific tokenizer and vocabulary for Languange Model to adapt to MIMIC-CXR dataset. We adapt LLaMA2 Vicuna-7B-v1.5 (Touvron et al. (2023)) as Language Model, as in LLaVa. The input image resolution is

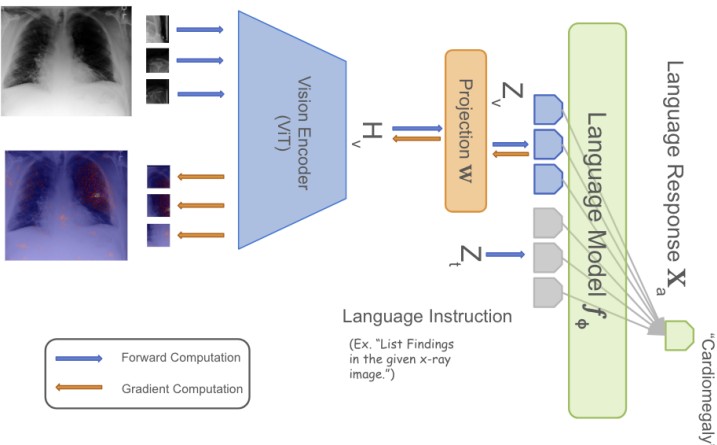

Figure 6: A schematic diagram showing saliency map generation in LVLM. A CXR image along with the instruction to list the abnormalities are given as input and saliency map is generated by taking gradient of the model output w.r.t the image

**MIMIC-CXR Instruction-Following Data Example**

**Reason:** Metastatic disease with known pulmonary metastases, presenting with fever and leukocytosis
**Impression:** Increasing left lung opacification which may reflect pneumonia superimposed on metastatic disease, although other etiologies such as lymphangitic pattern of metastatic spread could be considered. CT may be helpful to evaluate further if needed clinically.
**History:** Null
**Labels:** Lung Opacity, Pneumonia

**Query:** Describe the findings of the chest x-ray.
**Answer:** The lung volumes are low. The cardiac, mediastinal and hilar contours appear unchanged. There are a number of round nodular densities projecting over each upper lung, but more numerous and discretely visualized in the left upper lobe. There is a more hazy widespread opacity projecting over the left mid upper lung which could be compatible with a coinciding pneumonia. Pulmonary nodules in the left upper lobe are also not completely characterized on this study. There is no pleural effusion or pneumothorax. Post-operative changes are similar along the right chest wall.

Figure 7: An instance of Mimic-CXR Instruction following data used to train the model. The data contains CXR image along with information about the image in text form as shown in the figure

| Model | Rogue-1 | | Rogue-L | | BLEU-1 | | BLEU-4 | | Rad-graph | |
|---|---|---|---|---|---|---|---|---|---|---|
| | Orig | WA | Orig | WA | Orig | WA | Orig | WA | Orig | WA |
| Biomed-GPT | 13.19 | 06.60 | 10.05 | 05.52 | 08.20 | 04.02 | 0.99 | 0.50 | 06.32 | 02.71 |
| Rad-FM | 23.10 | 10.91 | 15.88 | 08.30 | 17.06 | 07.33 | 02.71 | 01.02 | 12.00 | 04.66 |
| LLaVA-Med | 21.97 | 16.22 | 14.61 | 11.91 | 15.09 | 10.57 | 01.20 | 00.92 | 10.03 | 09.45 |
| XrayGPT | 18.91 | 09.04 | 13.41 | 07.42 | 13.59 | 05.73 | 01.24 | 00.54 | 06.39 | 02.61 |
| LLaVA-CXR(MTL) | 32.49 | 31.25 | 22.04 | 21.33 | 25.02 | 24.75 | 04.61 | 04.20 | 18.91 | 18.83 |
| LLaVA-Rad | 12.76 | 08.77 | 10.64 | 07.70 | 00.94 | 00.42 | 00.10 | 00.05 | 02.47 | 01.74 |
| CheXagent | 33.21 | 25.51 | 24.04 | 18.01 | 26.69 | 19.66 | 07.08 | 03.07 | 18.81 | 13.58 |
| LLaVA-CXR(TACS) | 34.73 | 33.62 | 24.31 | 23.10 | 27.92 | 26.68 | 05.13 | 5.10 | 21.16 | 20.54 |

Table 2: Comparison of radiology report for CXR images from the primary and auxiliary task in MIMIC-CXR original and WhatsApp test images, generated by SOTA LVLM, LLaVA-CXR(full-MTL) and LLaVA-CXR(TACS). The results show that LLaVA-CXR(TACS) outperforms the baselines

| Model | Accuracy | | Precision | | Recall | | F1-Score | |
|---|---|---|---|---|---|---|---|---|
| | Orig | WA | Orig | WA | Orig | WA | Orig | WA |
| CheXagent | 53.85 | 53.12 | 30.81 | 30.22 | 43.01 | 43.14 | 35.85 | 35.50 |
| Rad-FM | 55.18 | 57.84 | 28.15 | 25.86 | 45.26 | 40.08 | 34.71 | 31.43 |
| LLaVA-CXR(MTL) | 36.50 | 36.20 | 24.56 | 24.50 | 74.72 | 74.01 | 36.93 | 36.79 |
| LLaVA-CXR(TACS) | 31.04 | 30.81 | 25.03 | 24.85 | 88.86 | 88.20 | 39.04 | 38.76 |

Table 3: Comparison of diagnostic labels generated for CXR images from the primary and auxiliary task in MIMIC-CXR original and WhatsApp test images, generated by SOTA LVLM, LLaVA-CXR(full-MTL) and LLaVA-CXR(TACS). The results show that LLaVA-CXR(TACS) outperforms the baselines

$518 \times 518$. We first pretrained the projection layer of LLAVa-CXR for 1 epoch and did end-to-end finetuning for 2 epochs using LoRa (Hu et al. (2022)). The learning rate for pre-training is $1e-3$ and finetuning is $1e-4$ and uses the cosine strategy. We use AdamW (Loshchilov & Hutter (2017)) optimizer with $\beta_1 = 0.9, \beta_2 = 0.99$ and weight decay of $0.01$. The training process utilizes cross-entropy loss, applied in an auto-regressive manner, to optimize the generation of text output.

### 6.4.2 ViT Training Details

The ViT model is trained using distributed training using optimizer AdamW, learning rate .003, weight decay 0.3, warmup-epochs set to 3, lr warm-up decay 0.033 and warm-up optimizer as Linear. The batch-size is 64 and model is trained for 30 epochs with early stopping.

### 6.4.3 WhatsApp Dataset Creation

MIMIC-CXR test dataset is compressed by WhatsApp in a fully automated manner. The challenges in creating this dataset is 1) WhatsApp sends images out-of-order in certain scenarios and 2) the metadata of images is lost after sending, and images are renamed to some random name by WhatsApp at the recipient's end. Since we do not have control over WhatsApp API, the image name is also sent along with the image. The steps involved in creating the CheXwhatsApp are given below:

- Preprocess the image by adding the image name to the image itself. This is done by adding a small white border of 30pixel $\times$ image-width at the bottom of the image and placing the image name in the white border. The preprocessed images are uploaded to an Amazon S3 bucket to be sent to the WhatsApp Business API.

- The pre-processed images are send by the WhatsApp business API one by one to a different WhatsApp account. This is established by using platforms for WhatsApp bulk-messaging. Such platforms are configured using a phone number enabled with WhatsApp and allow bulk-messaging of images and text.

| Atelectasis | Cardiomegaly | Consolidation | Edema | Enlarged Cardiomediastinum |
|---|---|---|---|---|
| -0.92 | -0.86 | -0.67 | -0.92 | -0.29 |
| Lung Lesion | Lung Opacity | No Finding | Pleural Effusion | Pleural Other |
| 0.99 | -0.47 | 0.98 | -0.96 | 0.99 |
| Pneumonia | Pneumothorax | Support Devices | | |
| 0.94 | 0.98 | -0.95 | | |

Table 4: TACS Score of primary task Fracture with auxiliary tasks in MIMIC-CXR evaluated using saliency maps generated from Vit-B/14

- The images received via WhatsApp are downloaded to an edge device, from where they are uploaded to a cloud service provider.
- The white border at the bottom of the images is cropped to read the image name via an OCR technique called Tesseract. The image name read via OCR is used to rename the image.

The original 5131 images in test dataset occupies 8.34 GB, while the corresponding WhatsApp compressed images occupies 1.22 GB

