# OpenReview forum: "TACS-Guided Self-Alignment of LVLMs for Explainable Chest X-ray Analysis"
_ICLR.cc/2026/Conference — ICLR 2026 Conference Withdrawn Submission_

### Official Review · Reviewer_CjUq · 2025-10-29

**Soundness:** 3
**Presentation:** 3
**Contribution:** 2
**Rating:** 4
**Confidence:** 5

**Summary:**

We propose Self-CXRAlign, a self-alignment framework that improves explainability robustness via MTL-driven supervised fine-tuning. The method ensures prediction and attribution consistency across clean and perturbed CXR inputs.

**Strengths:**

The paper is well written with a precise problem formulation. The proposed multi-task learning (MTL) framework for modeling inter-task relationships and identifying auxiliary tasks that avoid negative transfer is interesting and well motivated. The idea to use selected auxiliary tasks for self-alignment to improve LVLM robustness to compression artifacts addresses a practical failure mode for clinical imaging deployed over low-bandwidth platforms.

**Weaknesses:**

1. Motivation relies on assumptions about perturbations. It is unclear how the authors decide which perturbations (types and strengths) are “optimal” or realistic for the target deployment (e.g., WhatsApp compression). The paper would benefit from a clearer justification for the chosen corruptions and how representative they are of real-world transmission artifacts.
2. Training efficiency. The proposed approach requires training combinations of the primary task with each auxiliary task, this could be computationally expensive. More discussion or methods to mitigate training cost are needed.
3. The evaluation should report clinically meaningful metrics commonly used in the domain (e.g., CheXbert-F1 for label prediction when applicable) and more robustness analyses (e.g., performance vs corruption strength, explanation faithfulness).
4. Table3b seems missing, which is important to show the soundness of motivation for this work.

**Questions:**

What is the strategy to generate explanation heatmaps?

---

### Official Review · Reviewer_tgNJ · 2025-10-31

**Soundness:** 1
**Presentation:** 1
**Contribution:** 2
**Rating:** 2
**Confidence:** 3

**Summary:**

The paper introduces TACS, a metric to assess post-hoc saliency map consistency across non-compressed and compressed images. Using TACS, the authors propose Self-CXRAlign, a method to find which sub-tasks promote explanation consistency without the need
to train the model on all possible sets of sub-tasks, showing that Self-CXRAlign improves the report generation and OLS. Even though Self-CXRAlign seems promising, some methodological decisions and the lack of broader results raise concerns about the technique.

**Strengths:**

· Making the saliencies consistent between compressed and non-compressed images results in better performance for report generation, which is an interesting finding.
· The attribution vulnerability and TACS metrics introduced by the paper are valuable and can be used in other contexts.

**Weaknesses:**

· The method's primary goal is to make the saliency maps consistent across compressed and non-compressed images. Nevertheless, these saliencies are generated by only observing the gradients or the attention scores at different levels of the network. It is
unclear why the authors chose this approach instead of adapting the well-described methods in the literature, such as Grad-CAM, RISE, etc. References are also missing in this most important section of the work.

· Since the authors are using this method to generate the saliencies, some metric to evaluate the faithfulness of the explanations should have been performed, i.e, to check if the highlighted pixels are important for the model's final prediction using metrics such as
MoRF and LeRF.

· As observed in Figure 3a, the OLS score is relatively high for compressed and non-compressed images. This might indicate that the model performs similarly for both compressed and non-compressed images, or that the generated saliency maps perform
poorly.

· Figures 1,2, and 5 should be improved.

· The results are evaluated only when Fracture was selected as the primary task.

· The results are not sufficient to support the goal of the method.

· The work does not have a conclusion section.

**Questions:**

· Why did the authors decide to generate the saliency maps using the described method? How can the authors check that these maps are meaningful?

· In Section 3.3, the authors claimed that an attribution map for MTL does not exist in the literature, and then decided to take the average of the saliencies for each label as the final one. Can the authors clarify why it was done that way? Why not only have separated
saliencies for each label identified by the model?

· Section 4.1 uses a surrogate model (vision-transformer) to find the best tasks that optimize the consistency among the saliencies for the compressed vs non-compressed. With the identified tasks, the LVLM is then fine-tuned. How are the saliencies generated
for the surrogate model? Section 3.1 only defines how to generate the saliencies for the LVLM.

· Why is the PIP Score not reported in the results for Self-CXRAlign?

· Self-CXRAlign is fine-tuned using Fracture as the primary task, and the sub-tasks are identified using the method. Table 1 presents the results for the reports generated for the Fracture label. Did the authors also fine-tune the other methods/models similarly? This
would make this comparison fairer to the SOTA alternatives.

· Table 3 in the Appendix shows that the proposed method performs worse than the others in the accuracy and precision metrics. Why do the authors claim that it outperforms the baselines?

---

### Official Review · Reviewer_BmGJ · 2025-10-31

**Soundness:** 2
**Presentation:** 2
**Contribution:** 2
**Rating:** 2
**Confidence:** 4

**Summary:**

This paper tackles robustness and explainability in Large Vision-Language Models (LVLMs) for chest X-ray (CXR) analysis. The authors identify two main issues: (1) model instability, where LVLMs produce inconsistent predictions for original versus WhatsApp-compressed images, and (2) poor explainability due to Out-of-Lung Saliency (OLS), models attributing lung findings to irrelevant regions. To address these, they propose Self-CXRAlign, a self-alignment framework that uses Multi-Task Learning (MTL) and Supervised Fine-Tuning (SFT). The core idea is the Inter-Task Attribution Conflict Score (TACS), which guides the selection of auxiliary tasks by favoring those with negative attribution correlations (i.e., conflicting tasks). The authors claim this reduces attribution vulnerability and improves robustness, reporting up to 30% reduction in OLS and 45% improvement in report generation metrics compared to SOTA LVLMs.

**Strengths:**

- Real-world robustness to image perturbations (e.g., messaging-app compression) is an important and underexplored limitation in medical LVLM deployment.
- Formalizing Out-of-Lung Saliency (OLS) as a measurable failure mode is valuable for interpretability studies.
- Novel concept: The idea of using inter-task attribution conflicts (negative TACS) to guide multi-task training is theoretically interesting.
- The framework leverages existing model attributions rather than requiring additional manual labels, which is conceptually efficient.

**Weaknesses:**

Despite its novelty, the paper suffers from serious validation and evaluation issues.

- The authors train a LLaVA-CXR (full-MTL) model on all 14 tasks but never report its results on the full test set. Instead, they compare a TACS-guided model (9 tasks) against the 14-task models only on the 9-task subset, effectively comparing a specialist to a generalists. This looks unfair and invalidates the claim of a 45% improvement. The experiment should include full test results and fair baselines trained on equivalent subsets.
- The benefit of the TACS metric is conflated with that of the Self-CXRAlign SFT pipeline. Missing a crucial control experiment: retraining a SOTA baseline (e.g., LLaVA-Rad) on the 9-task subset identified by TACS. If this baseline also improves, the SFT procedure adds no value beyond TACS filtering
- The entire motivation that LVLMs fail on WhatsApp-compressed images is not empirically proven.
- To establish instability, the authors should have evaluated baseline LVLMs on original vs. compressed CXRs over multiple inference runs (e.g., 5–10 inferences) and reported per-abnormality average, variance, and instability rates. Robustness should also have been tested on multiple perturbation types (different JPEG quality levels). Without this, the motivation remains unclear
- The model explicitly drops five abnormalities (e.g., Pneumonia, Pneumothorax, Lung Lesion) that have positive TACS scores. This means the final model is blind to critical pathologies, which makes it unsafe for clinical deployment. The paper does not quantify this trade-off or explain how such omissions could be mitigated.
- TACS-guided SFT is tested only on the authors’ own LLaVA-CXR model and on one dataset (MIMIC-CXR). There is no demonstration that TACS generalizes to other datasets (e.g., CheXpert, PadChest), which limits its scientific impact.
- The paper does not report the sample counts or data distributions for the 14 abnormalities. Without this, we cannot tell whether TACS captures genuine attributional conflict or simply correlates with data imbalance.

**Questions:**

- Please provide the full evaluation of LLaVA-CXR (full-MTL) on all 14 tasks, including original and WhatsApp-compressed versions, and a comparison with other SOTA models on the full test dataset.
- What is the performance of the model on the five dropped tasks (Pneumonia, Pneumothorax, etc.)?
- How can a model that omits these critical findings be considered clinically valid?
- Why was a simple baseline retrained on the 9-task subset not included to isolate TACS’s effect?
- Please report training-sample counts per task and analyze their correlation with positive/negative TACS scores.

---

### Official Review · Reviewer_NPeS · 2025-11-02

**Soundness:** 1
**Presentation:** 1
**Contribution:** 1
**Rating:** 2
**Confidence:** 5

**Summary:**

The paper introduces Self-CXRAlign, a self-alignment framework designed to improve the robustness and explainability of large vision–language models (LVLMs) for chest X-ray analysis. The method employs multi-task learning–driven supervised fine-tuning (SFT) guided by a new metric, the Inter-Task Attribution Conflict Score (TACS), to reduce attribution conflicts. Experiments show that Self-CXRAlign improves stability and reduces OLS, supporting more reliable LVLM deployment in mHealth settings.

**Strengths:**

The paper addresses an important and practical problem in deploying LVLMs for medical imaging, emphasizing robustness and explainability in real-world clinical and mHealth settings.

**Weaknesses:**

The paper’s overall structure is poorly organized and many key experiments are placed in the supplementary materials, while Section 3 is overly lengthy and cluttered with unnecessary formulas, reducing readability and focus on the main contributions.
The Related Work section lacks sufficient discussion of existing research on explainability analysis, missing connections to prior interpretability frameworks in medical LVLMs.
In the experimental part, the authors mention constructing a WhatsApp-compressed dataset but fail to reference or compare with CVPR 2025’s “CheXwhatsApp” work, which directly addresses similar challenges. Moreover, the use of the same dataset name in the supplementary material requires clarification.
The use of saliency and attention maps as evidence for explainability is limited; these methods are not fully appropriate for evaluating LVLM interpretability. More rigorous validation involving clinical experts would be necessary to support the claims of improved explainability.

**Questions:**

1. Many key experiments are placed in the supplementary materials, and Section 3 contains lengthy formulaic derivations. Could the authors clarify why these details were not integrated into the main text and how they plan to improve the paper’s overall structure and readability?

2. The Related Work section lacks discussion of explainable analysis methods. How does the proposed framework relate to prior studies on interpretability in LVLMs, and what are its unique contributions in this context?

3. The paper mentions constructing a WhatsApp-compressed dataset but does not cite or compare with CVPR 2025’s “CheXwhatsApp” dataset. Could the authors explain the relationship between their dataset and that work, especially given the identical dataset name used in the supplementary materials?

4. Saliency and attention maps are used as the main explainability evidence. Could the authors justify why these visualizations are sufficient for assessing LVLM interpretability, and whether any clinical expert validation was conducted to support the conclusions?

---

### Note · Authors · 2025-11-27

I have read and agree with the venue's withdrawal policy on behalf of myself and my co-authors.